# Exceptional Evolution of a Squamous Odontogenic Tumor in the Jaw: Molecular Approach

**DOI:** 10.3390/ijms25179547

**Published:** 2024-09-02

**Authors:** Miguel Alonso-Juarranz, Oscar De La Sen, Pablo Pérez, Maria Aranzazu González-Corchón, Santiago Cabezas-Camarero, Melchor Saiz-Pardo, Jesus Viñas-Lopez, Lucia Recio-Poveda, Luisa María Botella, Farzin Falahat

**Affiliations:** 1Oral and Maxillofacial Surgery Service, Hospital Clínico San Carlos, 28040 Madrid, Spain; mialon09@ucm.es (M.A.-J.); odelasen@ucm.es (O.D.L.S.); pablo.perez@salud.marid.org (P.P.); mariaaranzazu.gonzalez@salud.madrid.org (M.A.G.-C.); santiago.cabezas@salud.madrid.org (S.C.-C.); melchor.saiz@salud.marid.org (M.S.-P.); 2Department of Surgery, Faculty of Medicine, Universidad Complutense, 28040 Madrid, Spain; 3Histopatology Service, Hospital Clínico San Carlos, 28040 Madrid, Spain; 4Oncology Service, Hospital Clínico San Carlos, 28040 Madrid, Spain; 5Secugen, Center for Biological, Research Margarita Salas, CSIC, 28040 Madrid, Spain; j.vinas@secugen.es; 6Department of Molecular Biomedicine, Center for Biological, Research Margarita Salas, CSIC, 28040 Madrid, Spain; luciarecio@hotmail.com; 7Rare Diseases Networking Biomedical Research Centre (CIBERER), 28029 Madrid, Spain

**Keywords:** squamous odontogenic tumor (SOT), rare cancers, malignization, personalized medicine

## Abstract

A squamous odontogenic tumor (SOT) is an epithelial locally benign neoplasia derived from the periodontium of the jaws. It is considered a lesion of low incidence. Predominantly, it affects the mandible, although both jaw bones may be involved. Here, we discuss the malignant clinical evolution of an SOT lesion in an 80-year-old female patient. The patient exhibited an expansive triangular lesion at the inferior right quadrant. Surgery was performed and an SOT was diagnosed (2019). Two years after, the lesion grew, and the analysis of the biopsy revealed SOT malignization with pleomorphic atypical squamous cells, characteristics of a squamous cell carcinoma (2021). Massive DNA sequencing of formalin-fixed–paraffin-embedded specimens of the initial and relapsed tumors indicated pathogenic mutations in *RET* and *POLE* genes in both tumors, loss of *ALK*, and gain of *CDKN1B* and *MAP2K* in the relapse. In addition, the clinical, radiographic, and microscopic features of this neoplasm are discussed and compared with those already published. The case presented contributes to the better understanding of this SOT tumor entity and to indicates its malignant evolution, together with its biological behavior and its histologic, clinical, and radiographic features. Also, it aims to stress the importance of deeper genetic analyses in rare diseases to uncover mutations that help to select a personalized treatment.

## 1. Introduction

A squamous odontogenic tumor (SOT) is classified by the World Health Organization as a locally benign epithelial tumor derived from the periodontium in the jaws [1,2,3,4]. It is considered a lesion of low incidence; approximately only around 170 cases have been reported in the literature to date [5,6,7]. Therefore, there is no casuistry wide enough to determine a precise clinical conduct.

Despite being a lesion that can appear at any age range, it predominates between the third and fifth decades of life, with an average age of 34–38 years old, and with a slight predisposition for the male gender [1,2,7].

The etiology and pathogenesis are not known. Most probably, they originate from the epithelial remnants of Malassez present in the periodontal ligament or in the adjacent mucous membrane, but this is not confirmed. In any case, these lesions are derived either from epithelial or mesenchymal elements, or both, and intrinsically related to tooth formation. There are known cases where the presentation affects various members of the same family, so it could be considered that there are predisposition mutations for this hereditary tendency [8,9,10]. They are located with similar frequency in both jaws, but the mandible is affected more often than the maxilla, with a preferential occurrence in the posterior premolar and molar area. Maxillary SOTs are described to be primarily present in the anterior area and appear to be more aggressive compared to those in the mandibular area. Multifocal lesions have been reported to be more frequent in these regions compared to other odontogenic tumors [4,11].

Usually, they are asymptomatic with slow intraosseous growth and may be associated with dental mobility, pain, and increased dental sensitivity, as well as mass effect [4,11,12]. When the lesion presents an expansive growth, a superficial ulceration could be observed.

Radiologically, an SOT appears as a unilocular radiolucent lesion with a triangular morphology and defined edges partially well delineated in the alveolar bone and sometimes resembling a periodontal defect. As indicated, they are associated with the roots of erupted, vital teeth and have a predilection for the anterior maxilla and the posterior mandible, being related to the displacement of the teeth [13].

Histologically, SOTs are similar to the cells and tissues that originate them. The tumors are locally benign neoplasms made up of nests of variable size and cords of uniform and well-differentiated squamous epithelial cells with occasional vacuolization and keratinization. They are surrounded by connective tissue [14].

Conservative treatment by curettage or enucleation of the lesion is usually curative, although more aggressive treatments such as block resection may sometimes be necessary [3,4,5]. The recurrences are not very frequent and, when they occur, are considered to be a consequence of incomplete removal of the injury [8,15]. Exceptionally, they can behave aggressively [4,5,12], tending to be more infiltrating and invading adjacent structures. In fact, and although SOTs are considered as benign lesions, the literature describes a few patients with aggressive behavior, acquiring the ability of invading the local tissues [11,12,15,16]. Despite this, malignant transformation has not been described beyond one case of concomitance of an SOT and squamous cell carcinoma in different areas of the mouth and one case of possible evolution to intraosseous squamous cell carcinoma [12,17].

With the aim of contributing to a better understanding of this entity in relation to its benign nature and its biological behavior, we describe here the case of a patient presenting an SOT that evolved into an invasive squamous cell carcinoma with tumor growth, requiring two surgeries in two years and oncologic treatment. This evolution of the disease is in contrast with a more classical evolution in this type of tumors. As far as we know, this is the first case of an SOT with this type of malignancy [12]. To find a putative triggering factor, massive DNA sequencing in original and relapsed tumor cells was performed, and mutations potentially implicated in the tumor malignization are described. In addition, the known literature on SOTs is critically reviewed. Knowledge and understanding of tumor pathophysiological features may help prognosis and improve the diagnostic accuracy and will allow the refinement of surgical and nonsurgical treatments and provide avenues for potential new therapies, particularly if malignization occurs. A better knowledge of this lesion is of great importance because it could provide information to improve its treatment. We want to outline the importance of personalized analysis in medicine, especially in inherited rare diseases, in the oncology field to uncover non-expected mutations.

## 2. Results

### 2.1. Clinical and Histological Manifestations of the Tumors

The case of study is an 80-year-old female that was referred to the Oral and Maxillofacial Surgery Service of the Hospital Universitario Clínico San Carlos in Madrid (Spain). She came with a leukoplakia in the right mandibular trigone without previous histology biopsy (September 2016). Despite the recommendation to perform a biopsy of the lesion, the patient did not come to the consultation for two years. When she returned (October 2018), she exhibited an outgrowth and friable lesion of 2 cm over the leucoplakia region. This lesion bulged through the vestibule and lingual surface. She did not report pain or hypoesthesia but did report grade 2 mobility of the tooth.

Given the patient’s symptoms, an orthopantomography (OPG) and a computed tomography (CT) were requested, showing a big multilocular radiolucent expansive lesion at the level of the right mandibular angle (Figure 1A,B). It extends from the distal root of tooth 46 to approximately 1 cm caudal to the mandibular notch, leaving a minimum basal bone thickness. No displacement or root resorption was observed.

Due to the clinical-radiological aspect, suspicious of malignancy, it was decided to perform an MRI and to take an incisional biopsy. The histological result showed a benign lesion with epithelial hyperplasia, hyperkeratosis, and parakeratosis with areas of fibrosis without being able to offer a clear diagnosis, a result that is repeated in multiple subsequent biopsies despite the aggressive behavior of the lesion. MRI revealed a tumor mass with mandibular infiltration and probable involvement of the masticatory space and deep parotid lobe, destroying the vestibular and lingual cortex and ipsilateral adenopathies at levels IB and IIA (Figure 2A,B). In the absence of a histological diagnosis of malignancy, it was decided to take cytological samples by fine-needle aspiration puncture (FNA) of the lymph nodes and take new serial biopsies in the operating room without detecting cells suspicious of malignancy in any of the samples.

During the following monitoring month, the patient developed a growth of the lesion, as well as hypoesthesia of the right inferior alveolar nerve and trismus. It was decided to request a PET-CT in which pathological uptake of fluorodeoxyglucose (FDG) was observed in the lesion with mandibular infiltration (SUVmax of 17.5) and probable ipsilateral lymph node involvement (SUVmax of 4.3) (Figure 3A,A′).

Given the unfavorable clinical evolution and the risk of fracture, a block resection of the lesion was recommended. Therefore, after the patient’s informed consent and under general anesthesia, a partial mandibulectomy associated with the extraction of tooth 46 with nerve preservation and placement of a reconstruction plate was performed. The patient presented a correct evolution and healing of the surgical site during the first postoperative month observed after OPG analysis (Figure 4A).

The histological result showed a tumor composed of tabs of mature squamous epithelium with focal cytokeratinization that grows with a pseudocystic pattern in a fibrous stroma without detecting neoplastic infiltration, compatible with a squamous odontogenic tumor (Figure 5A,A′).

After 6 months of follow-up, the patient reports a feeling of mandibular instability. Control imaging tests (OPG and CT) were requested, where it was possible to diagnose a recurrence of the intraosseous lesion that generated a lack of osseointegration of the osteosynthesis material. The option for reconstructive surgery with a microsurgical fibula flap was offered to the patient, but it was rejected.

Given the clinical situation of the patient, a surgical reintervention was decided to remove the osteosynthesis material. During the surgery, a significant increase in the tumor mass was evidenced, so the plate was removed, and a new tumor resection was performed without reconstruction (Figure 4B). The histological diagnosis continues being compatible with an SOT.

During follow-up, the patient remained stable. Several MRIs were performed during the following two years (June 2021), showing persistence of the lesion but no clear growth. After, during routine surveillance, an ulcerated lesion was observed in the right maxillary tuberosity, so it was decided to take a biopsy. The histological result was squamous cell carcinoma (Figure 5B,B′) with negative CPS (combined PD-L1 positive score) with higher expression of Ki67 and P53 and cell membrane delocalization of β-catenin (Appendix A).

PET-CT showed pathological uptake in the tumor bed of 12.3 SUVmax and 3.4 SUVmax in adenopathies of levels IB and IIA (Figure 6A). The case was discussed by the oncologic committee of professionals of the hospital, and it was decided to apply to the patient the ERBITAX chemotherapy scheme (paclitaxel + cetuximab). The patient received six cycles and had to be discontinued due to toxicity. It was therefore decided to prescribe cetuximab as a radiosensitizer in monotherapy and radiotherapy, 33 sessions of 70Gy in the tumor bed and 54Gy in the ipsilateral cervical lymph node chains. After this treatment, a control PET-CT scan was performed one month later, showing no tumor or lymph node uptake (Figure 6B). Currently (June 2024), the patient is still being followed up by our department with no signs of tumor recurrence.

### 2.2. Genetic Analysis of the Tumors

Table 1, Table 2 and Table 3 show the genetic variants found in the initial and the recurrence tumor. The exome analysis of the initial tumor revealed mutation in two genes, *RET* and *POLE*, which were also present in all the cells of the initial and in the tumor recurrence.

In fact, we have detected a variant c.2348A>C p.Asn783Thr (rs587778656, NM_020975.4) in the *RET* gene, which causes an amino acid change from asparagine to threonine at codon 783 within the intracellular tyrosine kinase domain. The value of the in silico predictor Combined Annotati on Dependent Depletion (CADD) is 24.8 (values above 20 are considered high probability of pathogenicity); the value of the in silico predictor Mendelian Clinically Applicable Pathogenicity (M-CAP) is 0.099. The allelic frequency for this variant in the gnomAD database is less than 0.01%, and the frequency at which this variant has been detected is 46%.

On the other hand, we have detected a t variant c.5174T>C p.Val1725Ala (rs1064796427, NM_006231.3) in the *POLE* gene, which causes an amino acid change from valine to alanine at codon 1725 within the DUF1744 domain in the C-terminal region. The frequency at which this variant has been detected is 55%. The value of the in silico predictor CADD is 24.2; the value of the in silico predictor M-CAP is 0.071; the allelic frequency for this variant in the gnomAD database is less than 0.01%. The frequency at which this variant has been detected is 55%.

In addition to variants in *RET* and *POLE* (possibly from germline) detected both in the primary tumor and the relapsed tumor, two pathogenic variants in the *NSD1* gene appeared in the primary tumor in a significant percentage of cells (15%) that were not detected in the relapsed tumor (if present, they should be in less than 5% of the cells) (Table 2). The variant in the *NSD1* gene, c.6050G>A p.Arg2017Gln (rs587784177, NM_022455.4) causes an amino acid change from arginine to glutamine in the 2017 codon of the protein. The other variant was c.2846del p.Pro949LeufsTer2 (NM_022455.4), which provokes the deletion of a cytosine in codon 949. Both variants have been identified with a frequency of 15% in the tumor sample and were not detected in the relapsed tumor (if present, they should be in less than 5% of the cells). The value of the in silico predictor M-CAP for the p.Arg2017Gln is 0.531.

The data from the biopsy obtained after the second surgery revealed a very high mutational rate in the relapsed tumor not present in the initial tumor. Variants in the *BRCA2* gene (p.Val1045AspfsTer5, COSV99061704) show a loss in copy number in the *ALK* gene and also a gain in copy number in the *CDKN1B* gene (cyclin-dependent kinase inhibitor 1B) and in the *MAP2K1* (mitogen-activated dual MAP kinase, MEK1) gene (Table 3).

## 3. Discussion

An SOT is histologically defined by the World Health Organization as a locally invasive neoplasm consisting of well-differentiated squamous epithelium clusters in a fibrous stroma [1]. In the literature, around 170 cases of SOTs have been described, both intraosseous or central and extraosseous or peripheral, with the former being much more common [3,6,10,15].

SOTs usually occur between the third and fifth decades of life, but cases have been reported at various ages [6,15]. The youngest patient described in the literature was a 9-year-old boy with a maxillary SOT treated with conservative surgery who experienced recurrence shortly after and required resection with margins [4,8]. Regarding gender, there is a slight preference for males (1.2:1), as well as greater involvement in black patients, who account for almost half of the cases described [6,15]. The case we present is unusual, as it involves a Caucasian woman in her eighth decade of life with an intraosseous SOT, which differs from an extraosseous SOT due to its growth towards the bone marrow.

SOTs affect the maxilla and mandible equally, with a predilection for the incisor-canine and molar areas [3,6,7]. While it is typically associated with teeth, cases of an SOT have been described in edentulous regions and even in relation to prosthetic materials [5,17,18,19]. Although it usually manifests as a single lesion, multicentric cases have been reported in the literature, including three within the same family, with slight variations in behavior [20,21,22,23]. The case presented here is consistent with the usual pattern, as it depends on the distal root of tooth 46 and presents a single focus.

Although an SOT can be symptomatic, it is often an incidental finding on imaging tests. When symptomatic, the most common symptoms include mobility and displacement of teeth, inflammation or erythema in the area, and erosion of the alveolar bone associated or not with pain, which can range from mild to moderate. These symptoms result from the expansive growth of the lesion [4,11,13]. It should be noteworthy that, in our patient’s case, the SOT initially presented as leukoplakia distal to tooth 46, without any associated symptoms, a presentation not previously described in the literature.

Radiologically, an SOT tends to present as a unilateral radiolucent image with a triangular morphology associated with the roots of adjacent teeth, without showing pathognomonic radiological signs [13,24]. However, our patient exhibited a radiolucent multilocular lesion as well as erosion and perforation of the bone cortex, both lingual and vestibular, consistent with other cases previously described [11].

Due to these characteristics, histological study is essential for accurate diagnosis. So, the histology of our case confirmed typical SOT architecture and composition, composed of cords and islands of well-differentiated nests of stratified squamous epithelium in the context of a fibrous stroma, benign in appearance and of variable size, embedded in mature connective tissue [1,25]. Likewise, the formation of keratin pearls, dystrophic calcification, and mucosal metaplasia of the squamous epithelium have been described in SOTs [9,26]. Although its origin is controversial, it is suspected that the intraosseous form derives from Malassez epithelial remnants, while the extraosseous form is derived from the dental lamina and gingival epithelium [20,27,28,29,30].

The differential diagnosis of an SOT includes periodontal disease as well as more aggressive lesions such as ameloblastoma, keratocyst, squamous cell carcinoma, or mucoepidermoid carcinoma [12,20,29]. Notably, squamous cell carcinoma and ameloblastoma share clinical and radiological similarities with SOTs but histologically present with atypia, cell mitosis, bone infiltration, or a perilesional cell palisade, respectively [20,30].

In our case, the initial SOT lesion evolved to a relapsed lesion suspicious of squamous cell carcinoma due to the clinical behavior suggestive of malignancy, including rapid and expansive growth, hypoesthesia of the inferior alveolar nerve, and associated lymphadenopathy. In addition, the histological analysis revealed that the epithelial islands showed atypical features suggestive of SCC with intense p53-Ki67-positive cells detected in the carcinoma areas. We also observed alterations in the membrane localization of β-catenin [31]. β-catenin is located at the cell membrane bound to E-cadherin and is involved in the organization of the epidermis, as well as playing a restrictive role in keratinocyte proliferation and migration. Many articles point out that loss of β-catenin at this location correlates with tumor grade and histological differentiation not only in OSSC but also in other carcinomas. The delocalization of β-catenin at the membrane is associated with increased cell motility and proliferation of epithelial cells [32,33].

In addition to the above observations, the exome analysis of the initial tumor revealed mutations in two genes, *RET* and *POLE*, which were also present in the cells of the initial and relapsed tumors, suggesting germline involvement. The *RET* gene encodes a receptor tyrosine kinase cataloged as a proto-oncogene associated with various tumors (ONIM:164761). The variant c.2348A>C p.Asn783Thr detected in the *RET* gene, which causes an amino acid change from asparagine to threonine at codon 783 within the intracellular tyrosine kinase domain, has a high probability of being pathogenic according to the value of the in silico predictor CADD. This variant is reported in ClinVar as of uncertain significance (ID: 135178), but there are no previous data in COSMIC. However, predictors indicate that this variant could be deleterious. Some authors suggest a possible influence of polymorphisms as modifiers of the risk of tumor development [34].

In addition, the variant c.5174T>C p.Val1725Ala, detected in the *POLE* gene, causes an amino acid change from valine to alanine at codon 1725 within the DUF1744 domain in the C-terminal region. The *POLE* gene encodes the catalytic subunit of DNA polymerase epsilon that is involved in DNA repair and chromosomal DNA replication. Alterations in this gene are associated with susceptibility to the development of tumors, mainly colorectal (OMIM:174762). In addition, it has been reported that cases of tumors that present alterations in *POLE* have a greater mutational load and microsatellite instability due to the reduced capacity for error correction (mismatch repair) during replication. The frequency at which this variant appears makes it plausible to be in the germline [35,36]. Also in this case, the value of the in silico predictor CADD of this variant as well as the value of the in silico predictor M-CAP make it plausible to be classified as pathogenic. In addition, the allelic frequency for this variant in the gnomAD database is less than 0.01% (only five cases reported in the South American population).

The combination of variants in the *RET* (rs587778656) and *POLE* (rs1064796427) genes could be involved in tumor development. The fact that both variants were found in a high frequency of cells and in both initial and relapse tumor sample points to the possibility that they were found in the germline and predisposed to tumor malignant development. Functional studies of the effect of these variants on tumor development would be necessary because they have not been previously reported in the literature.

Additionally, two pathogenic variants in the *NSD1* gene appeared in a significant percentage of cells in the primary but not in the relapsed tumor, suggesting tumor evolution or selection. Selection in the tumor may have caused them to disappear or to be found in less than 5% of the tumor cells at relapse.

The variant in the *NSD1* gene, c.6050G>A p.Arg2017Gln, causes an amino acid change from arginine to glutamine in the 2017 codon of the protein. The other variant, c.2846del p.Pro949LeufsTer2 (NM_022455.4), has also been detected in this gene, which causes the deletion of a cytosine in codon 949, which generates an alteration of the reading frame and a premature stop codon, giving rise to a truncated protein. Both variants have been identified in the tumor sample and were not detected in the relapsed tumor. The variant has been reported in COSMIC mainly associated with carcinomas (COSM1066299).

The obtained data in the sample from the second surgery revealed a very high mutational rate in the relapsed tumor. Apart from *RET* and *POLE*, the variant detected in the *BRCA2* gene (p.Val1045AspfsTer5) is clearly pathogenic for loss of function, affecting double-strand DNA repair and the genetic stability of the cellular subgroup that presents this alteration. This mutation was not present in the initial tumor [36,37].

In the initial tumor, no variation in the copy number was detected; however, the CNV in several genes appeared in the recurrence. The loss in copy number detected in the *ALK* gene may be associated with a genomic rearrangement not detected by the technique used. Alterations in *ALK* and *FGFR3*, present in the RTK/RAS/MAPK signaling pathway, have been reported in cutaneous squamous cell carcinomas; in particular, alterations in *ALK* have been described as drivers of metastases in squamous cell carcinomas to the lymph nodes [38,39]. The patient could benefit from treatment with *ALK* and MEK inhibitors (coinhibition) in case of relapse.

The gain in copy number detected in the *CDKN1B* gene may affect the function of the protein encoded by the gene, since it has been described in other types of tumors [37,40]. Alterations in other cyclin-dependent kinase inhibitors such as *CDKN2A* have been reported in cutaneous squamous cell carcinomas associated with alterations in *ALK* in tumors such as lung cancer [38].

Finally, the copy number gain detected in the *MAP2K1* gene could affect the function of the gene. Concomitant alterations in the *MAP2K1* and *FGFR3* genes, as it is in the present case, have been described in cases of cutaneous squamous cell carcinoma that presented a significant response to EGFR inhibitors (cetuximab) [36,38].

The benign behavior of an SOT and its low recurrence rate make conservative surgery the gold standard treatment, although more extensive surgery may be required in aggressive cases [1,2,4]. This treatment is especially indicated in mandibular lesions due to bone density, which seems to restrict tumor growth, as opposed to maxillary lesions, in which more aggressive surgery may be required given the porous characteristics of the bone [8,21,23,25,41]. In a few patients, as in our case, an SOT can present an aggressive behavior eroding the bone cortex, producing root resorption, hypesthesia, or recurrence [5,8,30], even having described the malignization of one case of an SOT in the right mandibular angle [12].

In about 170 published cases, only 5 of recurrence have been reported in the literature, 4 intraosseous and 1 extraosseous [2,4,6,7]. The recurrence developed over a short period of time (7–12 months) and, in all cases, the initial treatment was enucleation. The cure of the most aggressive cases was achieved after more extensive surgery including associated teeth [2,3].

In the case of aggressive behavior, extended surgery such as marginal or segmental resection with or without reconstruction may be the choice [2,4,8,21]. In our patient, despite a more extensive initial surgery than usual, based on a full-thickness ostectomy of the posterior mandibular sector, the facts of late diagnosis and aggressive characteristics of the primary tumor resulted in an early recurrence at six months. This behavior, although only one case has previously been described in the literature, could be explained by tumor infiltration of the perilesional soft tissues, which would make complete resection difficult [15].

Only one case of apparent malignancy of an SOT and one case of concomitance with squamous carcinoma have been described [12,17]. Therefore, our case is rather unique, and we want to stress the message of the importance of a genomic analysis together with pathological analysis to try to predict the evolution of this type of rare tumor and to apply and appropriate personalized treatment. In our case, the patient ended up developing squamous cell carcinoma on the SOT, requiring chemotherapy and radiotherapy treatment of the lesion, achieving complete remission.

## 4. Materials and Methods

### 4.1. Human Samples

Samples of an 80-year-old woman patient were processed for diagnostic pathology according to the usual procedure in the Hospital Clínico San Carlos. Once the pieces were obtained after tumor surgery, they were fixed in formalin and paraffin embedded as usual. Informed consent from the patient was obtained.

### 4.2. Magnetic Resonance Imaging

Magnetic resonance imaging (MRI) was performed using a SIEMENS, Magneton Verio, 3 Teslas, 32 channel antenna, analyzed by Numaris 4 Software (SIEMENS, Munich, Germany). For acquisition of all MRI images from the patient, a specific protocol was used, including axial T1, axial T2, axial FLAIR, T2* (gradient echo), diffusion-weighted (DW), axial T1 SPIR (fat suppression, contrast-enhanced), and 3D T1 SPACE (iso, contrast-enhanced).

### 4.3. Positron Emission Tomography-Computed Tomography

For standard 18F-FDG PET/CT, a blood glucose level of less than 200 mg/dL was administered at the time of 18F-FDG injection with an uptake time of 55–70 min. The patient was scanned from the skull base to clavicles with the arms down and then from the skull base to mid thighs with the arms up. The dedicated head and neck imaging protocol consists of 3D PET scans with 5 min for each bed position, a 30 cm FOV, and 256 × 256 matrix for improved resolution and sensitivity to detect small neck node metastases.

### 4.4. Histopathological Analysis

Histologic sections (5 µm) of the surgical pieces were mounted on slides with electrostatic charge for general structure (Superfrost Plus, Thermo Fisher, Waltham, MA, USA). Sections were then deparaffinized (15 min in xilol) and stained with hematoxylin/eosin (H/E) for histopathological analysis or subjected to immunohistochemistry. For immunohistochemistry (IHC), endogenous peroxidase was inhibited with 3% hydrogen peroxide (Panreac, Barcelona, Spain) in methanol. The antigen unmasking was performed by immersing them in citrate buffer at pH 6 (0.25% citric acid and 0.038% sodium citrate in water) for 10 min in a pressure cooker. When the samples had tempered, blocking was carried out with non-immune serum (Dako, Agilent Technologies, Santa Clara, CA, USA) for 1 h at room temperature, followed by overnight incubation at 4 °C with the primary antibody (Ki67, P53, and β-catenin) (Abcam, Cambridge, UK). Thereafter, the sections were incubated with the secondary antibody conjugated to streptavidin-peroxidase (Cell Signaling Technology, Inc., Danvers, MA, USA) for 30 min at room temperature. Color was developed by 3-amino-9-ethylcarbazole solution as chromogen (DAB, Vector Laboratories, Burlingame, CA, USA), and hematoxylin counterstaining was performed following the manufacturer’s protocol. Finally, the sections were dehydrated in increasing series of alcohol and mounted with DePeX (Serva, Heidelberg, Germany).

### 4.5. Molecular Analysis: DNA Total Extraction and Sequencing

Total DNA was extracted from 20 slices (5 µm) of paraffin-embedded tumor pieces using the QIAamp Mini Kit (Qiagen, Düsseldorf, Germany) and following the manufacturer’s indications. The panel used for the preparation of the library was designed using SureSelectXT technology (Agilent Technologies, Santa Clara, CA, USA), aimed at capturing the exons of the genes of clinical interest and the flanking splicing regions (5–20 bp). Sequencing of the library was performed on a next-generation mass sequencer, NovaSeq 6000 SystemTM (Illumina, San Diego, CA, USA). The tumor samples were sequenced with a reading depth of 500x. The sequences obtained were aligned against the reference genome (GRCh38/hg38) and filtered according to specific quality criteria. Subsequently, they were analyzed for the identification of genetic variants included in exonic regions or splicing regions (at least 5 bp), including missense or nonsense mutations, synonymous mutations, indels, small insertions, or deletions found at a higher allele frequency (>30% of germline reads and >5% of tumor sample reads). Both processes were carried out using the DRAGEN^TM^ BioIT Platform software (Illumina, version 07.021.510.3.5.7). The identified variants were filtered and narrowed down to the study genes using the bcf-tools view tool (version 1.19) [42,43].

Variant annotation was performed using the freely available online platform ANNOVAR (WGLAB, PA, USA), which compiles the main databases, major databases such as ClinVar (with specific information on variants associated with a known genotype), and databases (October 2023) of population frequency data—dbSNP, gnomAD (Genome Aggregation Database), 1000 Genome Project, or NHLBI-ESP 6500 exons.

The pathogenicity of the variants was also estimated using CADD and the addition of selected prediction systems included in the dbNSFP database (SIFT, PolyPhen2, Mutation Taster, Mutation Assessor, LRT, FATHMM, and MetaSVM) for missense mutations. The conservation of nucleotide position has been evaluated according to the UCSC score ranges for the PhyloP tool (version 3.19 compatible with human genome GRCh38/hg38).

Finally, the association of the identified variants with OMIM syndromes has been evaluated. The nomenclature and classification of the variants is based on the guidelines of the Human Genome Variation Society (HGVS) (http://varnomen.hgvs.org/, accessed on 7 May 2024) and the American College of Medical Genetics and Genomics (ACMCG). The analysis of copy number variations (CNVs) is a screening performed with the established parameters based on a set of control samples with the DRAGEN software (version 07.021.572.3.6.3). This algorithm allows the identification of non-recurrent CNVs associated with the patient’s phenotype following the quality criteria.

The variants described in population databases classified as polymorphisms or single nucleotide polymorphisms (SNPs) and/or rare variants (population frequency less than 1%), mostly reported in the databases consulted as benign or probably benign, were not taken into consideration.

## 5. Conclusions

The present work shows the importance of personalized medicine in the case of an SOT. An SOT is a rare benign odontogenic epithelial neoplasm, usually asymptomatic, that presents a slow growth. It does not show a characteristic radiographic appearance, so it can be clinically confused with other odontogenic lesions. The usual treatment consists of conservative surgery based on curettage and enucleation of the lesion given its low rate of recurrence. However, the SOT in our patient presented an aggressive behavior, despite a more extensive initial surgery than usual, that resulted in an early recurrence at six months with SCC histological characteristics. In addition to that, the genetic analysis indicated mutations in *RET* and *POLE* genes (at the initial and in the relapse tumors) and variants in *NSD1* (at the initial) and *BRCA2* (in the relapsed), as well as CVNs in *ALK*, *CDKN1B*, and *MAP2K1* genes. All these genetic alterations could cause the abnormal severity in the course of this disease and, consequently, the disease is sped up and the patient underwent multiples biopsies. In the end, the patient developed an epidermoid carcinoma, which had to be treated oncologically. Thus, we have provided an explanation why this SOT patient has an accelerated growth and highlight the importance of exhaustive genetic analyses to uncover mutations in additional genes. This can be applied for other SOT cases to individualize the treatment of each patient based on their clinical and radiological behavior. An SOT is a rare entity that must be taken into account in the differential diagnosis of maxillary and mandibular tumors and, on occasion, can malign.

## Figures and Tables

**Figure 1 ijms-25-09547-f001:**
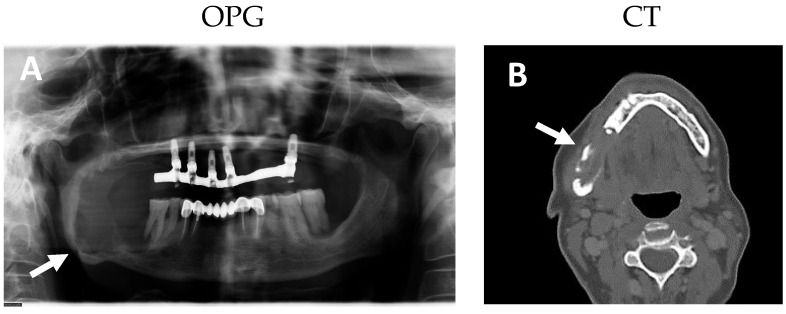
Pre-surgical orthopantomography (OPG), computed axial tomography (CT). An expansive and osteolytic lesion in the right mandibular angle lesion with a basilar remnant of about 1 cm in diameter is observed in the OPT image ((**A**), arrow). The CT indicates that the lesion affects both mandibular cortices ranging from the right mandibular angle to tooth 45 ((**B**), arrow).

**Figure 2 ijms-25-09547-f002:**
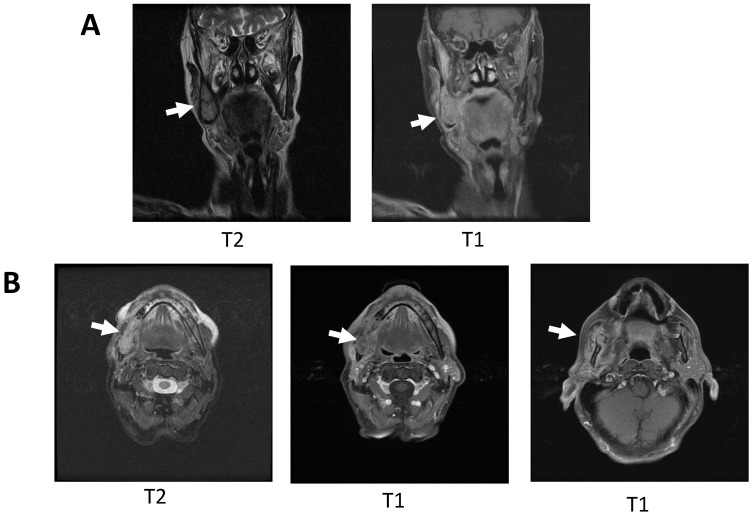
MRI obtained prior to surgery. (**A**): Coronal cuts (T2 and T1) with expansive bone lesion of the right mandibular angle (arrows). (**B**): Axial cuts (T2, T1, and T1) showing expansive bone lesion of the right mandibular angle with loss of continuity of the lingual cortex and involvement of adjacent soft tissues (arrows).

**Figure 3 ijms-25-09547-f003:**
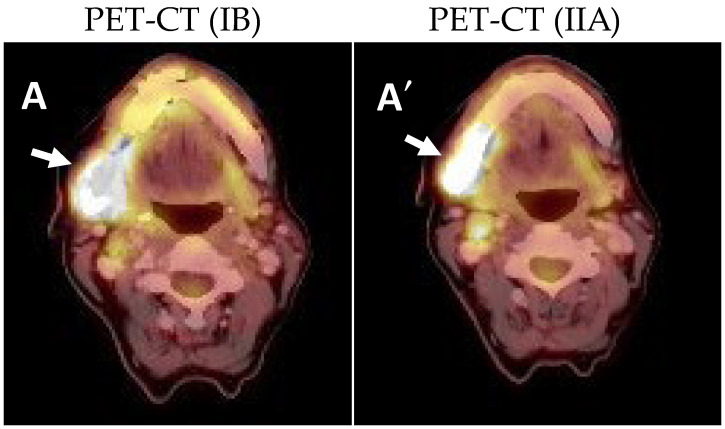
Pre-surgical PET-CT showing the uptake in the right mandibular angle of SUVmax 17.5 with adenopathy at levels IB (**A**) and IIA (**A′**) with uptake of up to SUVmax 4.3 (arrows).

**Figure 4 ijms-25-09547-f004:**
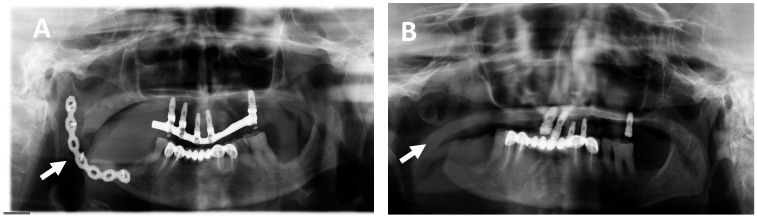
Post-surgical orthopantomographies (OPG) image showing the reconstruction plate of 2.0 mm thickness before (15 January 2019) (**A**) and after removal of the plate (23 March 2021) (**B**).

**Figure 5 ijms-25-09547-f005:**
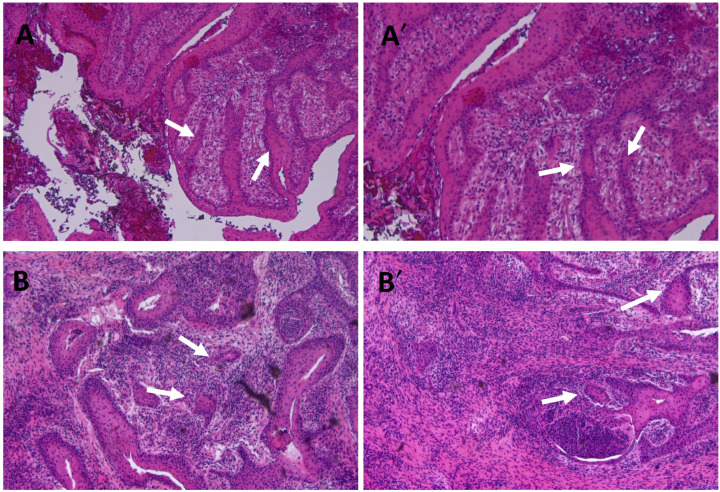
Hematoxilin eosin staining of the biopsy obtained from the lesion located in the right upper jaw. The lesion corresponded to an odontogenic squamous tumor characterized by the formation of variably sized nests and cords of uniform, benign-appearing, squamous epithelium with occasional vacuolization and keratinization (arrows) ((**A**): 10× and (**A′**) 20×). Histopathological analysis of the biopsy obtained from the recurrent lesion corresponding to the mucosa infiltrated by squamous cell carcinoma. The mucosa shows infiltration of the stroma by nests of moderately squamous differentiation cells with little nuclear and cellular atypia, with cytoplasmic keratinization that focally shows cystic areas (arrows). No vascular or perineural invasion is identified ((**B**): 10× and (**B′**): 20×).

**Figure 6 ijms-25-09547-f006:**
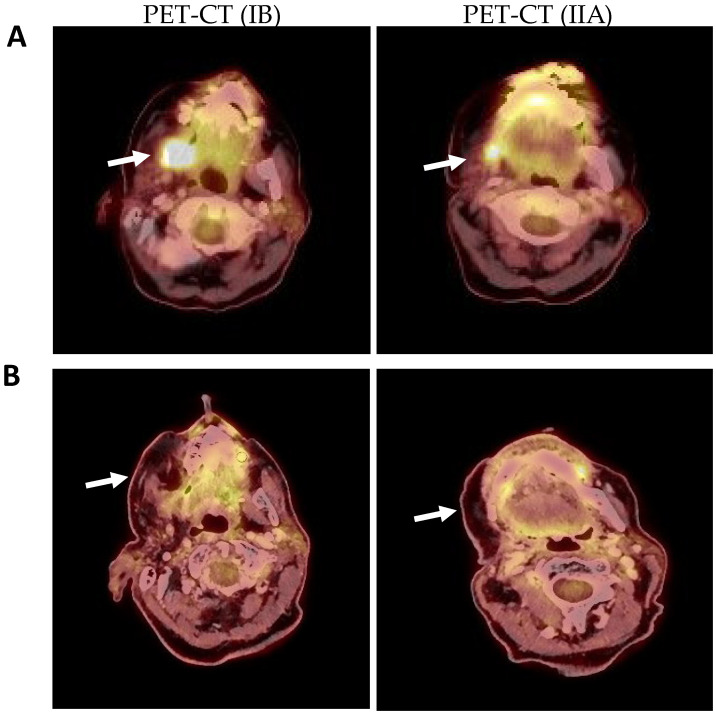
(**A**): Post-surgical PET-CT (August 2021) prior to oncologic pretreatment with uptake in the right maxillary tuberosity of SUVmax 12.3 and in lymphadenopathy IB and IIA of SUVmax 3.2. (arrows). (**B**): PET-CT obtained after treatment with radiotherapy and chemotherapy (24 January 2022) where an absence of uptake is observed both in the right maxillary tuberosity and in previously affected adenopathic levels (arrows).

**Table 1 ijms-25-09547-t001:** Results obtained from the massive sequencing study (more than 500 readings per area, clinical exome) to identify the genomic variants in the genes in paraffinized tumor tissue sample (odontogenic tumor) after relapse. The variants in the indicated exons of *RET*, *POLE*, *BRCA2,* and *FGFR3* were detected. (*): variants detected in *RET* and *POLE* possibly come from germline. Frec. genotype alt.: frequency of genotype alteration.

Gene	Region	HGVSC	Frec. Genotype Alt.	Ref. ID
*RET* *	Exon 13	NM_020975.4:c.2348A>C, p.Asn783thr	46%	rs587778656
*POLE* *	Exon 39	NM_006231.3:c.5174T>C, p.Val1725Ala	55%	rs1064796427
*BRCA2*	Exon 11	NM_0000059.3:c.3134insAT, p.Val1045AspfsTer5	9.5%	-
*FGFR3*	Exon 11	NM_001163213.1:c.1534>A, p.Leu512Met	2.6%	-

**Table 2 ijms-25-09547-t002:** Results obtained from the massive sequencing study (more than 500 readings per area, clinical exome) to identify the genomic variants in DNA from paraffinized tumor tissue sample (odontogenic tumor) in the first surgery. In addition to variants in *RET* and *POLE* indicated in Table 1, variants in the indicated exons of *NSD 1* are shown. (*): variants detected in *RET* and *POLE* possibly come from germline Frec. genotype alt.: frequency of genotype alteration.

Gene	Region	HGVSC	Frec. Genotype Alt.	Ref. ID
*RET* *	Exon 13	NM_020975.4:c.2348A>C, p.Asn783thr	46%	rs587778656
*POLE* *	Exon 39	NM_006231.3:c.5174T>C, p.Val1725Ala	55%	rs1064796427
*NSD1*	Exon 5	NM_022455.4c.2846del, p.Pro949LeufsTer2	15%	-
*NSD1*	Exon 20	NM_022455.4c.6050G>A, p.Arg2017Gln	15%	rs587784177

**Table 3 ijms-25-09547-t003:** Results obtained from the copy number variation (CVN) sequencing of the DNA from the paraffin tissue sample corresponding to the odontogenic lesion after relapse. Loss in copy number of the *ALK* gene and gain in the *CDKN1B* and *MAP2K* genes have been detected.

Gene	Citoband	Region	Consequence Alteration
*ALK*	2p23.2-p23.1	chr2:29694813-29920685	DEL:CNV-*LOSS*
*CDKN1B*	12p13.1	chr12:12717778-12719004	DUP:CNV-*GAIN*
*MAP2K1*	15q22.31	chr15:66387328-66490677	DUP:CNV-*GAIN*

## Data Availability

The original contributions presented in the study are included in the article and Appendix A, further inquiries can be directed to the corresponding authors.

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
