# Peer review of "Exceptional Evolution of a Squamous Odontogenic Tumor in the Jaw: Molecular Approach"

_ijms, 2024, doi:10.3390/ijms25179547_

Round 1

Reviewer 1 Report

Comments and Suggestions for Authors

In this article, the authors describe an evolution of a squamous ondontogenic tumor (SOT) in the jaw as a case study in an 80 years old female patient.

General comment:

Given the low incidence of SOT, this report is valuable and important to share. The methodology used to describe the case is adequate; namely, the authors used imaging (MRI and PET/CT), histopathological analysis (including immunohistochemistry) and next-generation sequencing. However, there are several points that should be improved.

Specific points:

Introduction is missing.

The authors could briefly add why (the rationale) did they study the localization of beta-catenin.

Figures 1, 2 and 5: The authors could use arrows to emphasize the areas of interest on the image.

Tables 1 and 2: The authors should explain what the abbreviation 'Frec. genotipe alt' means.

Lines 326-327: The sentence does not seem to be complete.

Author Response

Comments and Suggestions for Authors

In this article, the authors describe an evolution of a squamous ondontogenic tumor (SOT) in the jaw as a case study in an 80 years old female patient.

General comment:

Given the low incidence of SOT, this report is valuable and important to share. The methodology used to describe the case is adequate; namely, the authors used imaging (MRI and PET/CT), histopathological analysis (including immunohistochemistry) and next-generation sequencing. However, there are several points that should be improved.

We thank the reviewer for the comments and suggestions which have allowed us to improve the manuscript. All the changes suggested have been inplemented in the MS and are highlighted in red or yellow.

Specific points:

Introduction is missing.

We apologize, and feel very sorry. We had extracted the introduction to modify some sentences before  the submission of  the manuscript,, unfortunately we forgot to paste it again. Please, find it incorporated in the MS as it should be.

The authors could briefly add why (the rationale) did they study the localization of beta-catenin.

Thank you very much for your observation. We have added the following explanation in the article (Discussion section, lines 429-434).

"β-catenin is located at the cell membrane bound to E-cadherin and is involved in the organisation of the epidermis, as well as playing a restrictive role in keratinocyte proliferation and migration. Many articles point out that loss of β-catenin at this location correlates with tumour grade and histological differentiation, not only in OSSC, but also in other carcinomas. The delocalisation of β-catenin at the membrane is associated with increased cell motility and proliferation of epithelial cells".

This is why we have used this marker to better determine the undesirable evolution of the SOT described in our MS.

We have included two references supporting the role of β-catenin in oral cancer. 

Figures 1, 2 and 5: The authors could use arrows to emphasize the areas of interest on the image.

Thank you for the observation, we have added arrows in the areas of interest (Figures 1, 2, 5 and 6) as suggested.

Tables 1 and 2: The authors should explain what the abbreviation 'Frec. genotipe alt' means.

The "Frec. genotipe alt" has been written in long to explain the meaning: Frequency of Genotype Alteration.

Lines 326-327: The sentence does not seem to be complete.

The sentence has been revised and completed.

Reviewer 2 Report

Comments and Suggestions for Authors

In this manuscript, the authors discussed on the malignant clinical evolution of a SOT lesion in an 80 years old female patient.

1. The entire introduction section appears to be missing and seems to be taken directly from a template. This is a significant issue that should not occur in scientific publications.

2. Since the manuscript discusses only a single patient, it does not include any quantitative analysis with statistical confidence. Therefore, it should be considered as a case report rather than a research article.

3. The uniqueness of the case needs to be highlighted.

Comments on the Quality of English Language

/

Author Response

Comments and Suggestions for Authors

In this manuscript, the authors discussed on the malignant clinical evolution of a SOT lesion in an 80 years old female patient.

Thank you for your revision and suggestions. All the changes have been marked in the text, either in red when they were typos or small changes to improve English, or in yellow for larger changes and additions.

The entire introduction section appears to be missing and seems to be taken directly from a template. This is a significant issue that should not occur in scientific publications.

We apologize and feel  very sorry. We had withdrawn the introduction to modify some sentences before  the submission of the manuscript and unfortunately we forgot to paste it again. Please, find it attached as it should be.

Since the manuscript discusses only a single patient, it does not include any quantitative analysis with statistical confidence. Therefore, it should be considered as a case report rather than a research article.

We agree; the manuscript should be considered as a case report.

The uniqueness of the case needs to be highlighted.

Thank you once more.  To highlight the uniqueness of the case we have included the following paragraph at the end of the discussion:

"Therefore, our case is rather unique, and we want to stress the message of the importance of a genomic analysis together with pathological analysis to try to predict the evolution of this type of rare tumors, and to apply and appropriate personalized treatment".